# Effect of Increased Powder–Binder Adhesion by Backbone Grafting on the Properties of Feedstocks for Ceramic Injection Molding

**DOI:** 10.3390/polym14173653

**Published:** 2022-09-02

**Authors:** Laleh Ghasemi-Mobarakeh, Santiago Cano, Vahid Momeni, Dongyan Liu, Ivica Duretek, Gisbert Riess, Christian Kukla, Clemens Holzer

**Affiliations:** 1Department of Textile Engineering, Isfahan University of Technology, Isfahan 84156-83111, Iran; 2Polymer Processing, Montanuniversitaet Leoben, 8700 Leoben, Austria; 3Institute of Metal Research (IMR), Chinese Academy of Sciences, Shenyang 110016, China; 4Chemistry of Polymeric Materials, Montanuniversitaet Leoben, 8700 Leoben, Austria; 5Industrial Liaison Department, Montanuniversitaet Leoben, 8700 Leoben, Austria

**Keywords:** ceramic injection molding, feedstock, acrylic acid grafted high density polyethylene, paraffin wax, high density polyethylene, binder

## Abstract

The good interaction between the ceramic powder and the binder system is vital for ceramic injection molding and prevents the phase separation during processing. Due to the non-polar structure of polyolefins such as high-density polyethylene (HDPE) and the polar surface of ceramics such as zirconia, there is not appropriate adhesion between them. In this study, the effect of adding high-density polyethylene grafted with acrylic acid (AAHDPE), with high polarity and strong adhesion to the powder, on the rheological, thermal and chemical properties of polymer composites highly filled with zirconia and feedstocks was evaluated. To gain a deeper understanding of the effect of each component, formulations containing different amounts of HDPE and or AAHDPE, zirconia and paraffin wax (PW) were prepared. Attenuated total reflection spectroscopy (ATR), scanning electron microscopy (SEM), differential scanning calorimetry (DSC) and rotational and capillary rheology were used for the characterization of the different formulations. The ATR analysis revealed the formation of hydrogen bonds between the hydroxyl groups on the zirconia surface and AAHDPE. The improved powder-binder adhesion in the formulations with more AAHDPE resulted in a better powder dispersion and homogeneous mixtures, as observed by SEM. DSC results revealed that the addition of AAHDPE, PW and zirconia effect the melting and crystallization temperature and crystallinity of the binder, the polymer-filled system and feedstocks. The better powder--binder adhesion and powder dispersion effectively decreased the viscosity of the highly filled polymer composites and feedstocks with AAHDPE; this showed the potential of grafted polymers as binders for ceramic injection molding.

## 1. Introduction

Ceramic injection molding (CIM) is an established, accurate, cost-effective and efficient technology for the production of ceramic parts with complex geometries, with increasing interest [1,2,3]. The CIM technology involves the mixing of ceramic powders with polymeric binders and the shaping of a uniform and homogeneous mixture, named feedstock, into a mold cavity during the injection molding process. After injection molding, the debinding step is performed to remove the polymeric binders; then, the ceramic components are sintered to reach high density [3,4,5]. Loading high amounts of solids is a vital issue for the injection molding and the appropriate selection of ceramic/binder systems could be beneficial to provide high solid loading [4]. Polymeric binders have a vital role in the properties and successful fabrication of the final ceramic components; moreover, they act as a carrier for shaping and holding the ceramic powders together until the debinding. Thus, the design of a suitable binder is critical for the success of the process [3,6]. The basic purpose of binders is to assist in the shaping of the component during injection molding and to provide strength to the shaped component [3,7]. The properties of the binder effect the strength of the final ceramic components and the debinding efficiency [8]. Since it is not possible to achieve all the requirements optimally with only one polymer in the binder, multicomponent binders are preferred. Binders are usually mixtures of various polymers, with additives such as dispersants, stabilizers and plasticizers [2,7,9]. Plasticizers usually decrease the viscosity and improve the ductility; polymers are used to obtain a cohesive force between particles to keep the structural integrity after the main component removal in the debinding process; and additives are used to disperse particles and to avoid particle agglomeration [7]. Waxes such as paraffin wax (PW) are used as a main binder component as they provide high fluidity in the molten state; this is due to their low molecular weight and having relatively low shrinkage (0.05–1.11% shrinkage for wax vs. 1.5–4% and 1–3% shrinkage for HDPE and polypropylene, respectively) [3,10,11]. The waxes are usually combined with polyolefins, such as high-density polyethylene (HDPE) or polypropylene (PP); these act as a backbone and provide strength and stiffness to the parts [3,7]. Binders and ceramic powders require good adhesion to obtain homogeneity in the mixture and to prevent ceramic aggregation and phase separation through processing [12,13]. Good adhesion and dispersion of the ceramic powder are vital to successfully process by high-speed and high-pressure injection molding, without the separation of the powder and binder system [4]. Non-polar polymers such as polyolefins cannot have good interactions with the polar surface of the ceramic powder; this is due to the low adhesion and the lack of specific chemical interaction between the filler and polymer [13,14]. Different methods have been employed to improve the adhesion between the polymer and the ceramics [5]. As ceramic powders have high agglomeration ability, polar dispersants such as stearic acid are usually included in the binder to control ceramic powder dispersion [3,5,12,15]. Including polymers with polar groups as compatibilizers or interfacial agents in the binder formulation is beneficial to assist the interfacial interactions between the powder and the matrix [16,17,18]. Wongpanit et al. [18] investigated the role of the acid-grafted high-density polyethylene (AAHDPE) ratio (0, 5, 10, 25 vol%) in a high-density polyethylene (HDPE) based binder on the properties of fabricated parts after injection molding and debinding. Their results revealed that AAHDPE acts as a dispersant; it improved the adhesion between the binder and powder, increased the strength of the green parts and reduced the shrinkage during injection molding. In our previous study, zirconia was mixed with AAHDPE or HDPE. The results showed an improvement of the powder dispersion and the mechanical properties of the zirconia-polymer mixture; this was due to an enhancement in the adhesion between zirconia and AAHDPE [13]. Although many binder and polymer-filled formulas have been established, there are only a few studies focused on the use of grafted polyolefins as binders; furthermore, the effect of these polymers on the feedstock structure and properties is not yet fully understood.

This study was aimed to investigate the effect of the addition of AAHDPE on the properties of a zirconia feedstock, with PW as the main binder component and HDPE as the polymeric component. In order to do so, different formulations with increasing complexity containing HDPE and or AAHDPE, PW and zirconia were developed; the morphology, rheological, chemical and thermal properties of these formulations were evaluated and compared. To the best of our knowledge, no deep report on the formulations with different constitutes of PW as a plasticizer, AAHDPE as a compatibilizer and zirconia as a filler has previously been published.

## 2. Materials and Methods

### 2.1. Materials

High-density polyethylene (HDPE) and acrylic acid-grafted high-density polyethylene (AAHDPE) were obtained from Borealis AG (Vienna, Austria) and BYK-Chemie GmbH (Wesel, Germany), respectively; they were used as a polymeric system. More detailed characteristics and information about HDPE and AAHDPE were stated in our previous study [13]. The paraffin wax (PW), Sasolwax 6403 (PW, Sasol Limited, Johannesburg, South Africa), was used during this study. Yttria stabilized tetragonal zirconia with an average particle size of 0.02–2000 μm; a specific surface area of 0.192 m^2^/g was obtained from Treibacher Industrie AG, (Althofen, Austria) and was used as filler in the formulation of the ceramic feedstock.

The different combinations of two polymers (HDPE, AAHDPE), zirconia and PW prepared during this study are listed in Table 1. With formulations A, the properties of the HDPE, AAHDPE and the combination of them were tested. Formulations B examined the combinations of the two types of polyethylene with PW in a 50/50 volume ratio. The adhesion of HDPE and AAHDPE with zirconia was tested with formulations C; this was with only 30 vol% as no more powder could be filled into the polymers. In formulations D, the mixtures of polyethylene and PW were filled with 30 vol% of zirconia to evaluate the effect of PW on the properties by comparison with series C. Finally, formulations E were included as examples of typical feedstocks for ceramic injection molding, with 50 vol% of powder. To designate the different formulations in each type of mixture, the fraction of polyethylene corresponding to AAHDPE was used.

### 2.2. Methods

#### 2.2.1. Preparation of the Compounds and Feedstocks

Different compounds were processed in a kneader with a mixing chamber with a volume of 38 cm^3^ (Plasti-Corder PL2000, Brabender GmbH & Co. KG, Duisburg, Germany). Mixing was carried out at 160 °C and 60 rpm. For the compounding, at first the polymers and PW were fed into the chamber; then, after 3 min the powder was added in 5 batches with 5 min after each addition to ensure the homogenization of the mixture and the stabilization of the torque; after filling, the mixing continued up to a total time of 45 min to facilitate a proper dispersion and homogenization. The molten compounds were extracted from the chamber and cooled down to room temperature. The polymeric system without zirconia (groups A and B) was prepared using the same method to prevent the effect of the thermo-mechanical processing step.

Many trials were conducted for the preparation of feedstock E0. However, due to the low interaction of zirconia and HDPE and the high amounts of zirconia in this formulation, this feedstock failed to prepare. Moreover, the formulations Ax and Cx could not be prepared due to the very low viscosity of PW in a molten state; this led to leakage in the mixing chamber.

To prepare the granules, the cutting mill Retsch SM200 (Retsch GmbH, Haan, Germany) was used.

#### 2.2.2. Preparation of Compression Molded Plates

To prepare the plate samples suited for doing rotational rheology, the hydraulic vacuum press P200PV (Dr. Collin GmbH, Maitenbeth, Germany) with the program stated in Table 2 was used. Plates were fabricated using the granules placed in a steel frame with a thickness of 2 mm and a diameter of 25 mm; and covered with flat polytetrafluoroethylene (PTFE) plates.

#### 2.2.3. Attenuated Total Reflection Spectroscopy

The infrared absorption spectra of different samples were obtained by attenuated total reflection spectroscopy (ATR). ATR spectroscopy was carried out using a Vertex 70 spectrometer (Bruker, Ettlingen, Germany) at room temperature over a range of 4000–600 cm^−1^ with a resolution of 2 cm^−1^.

#### 2.2.4. Morphology Analysis

The morphology of the polymer-filled materials (groups C and D) and feedstocks (group E) extruded in the capillary rheometer was studied by scanning electron microscopy (SEM, JSM-6301F, JEOL Ltd., Tokyo, Japan). The samples were coated with gold prior to SEM observation; and an operation voltage of 20 kV and secondary electron mode were selected for the observation of samples with SEM.

#### 2.2.5. Differential Scanning Calorimetry (DSC)

Thermal properties of the different samples were measured by differential scanning calorimetry (DSC 1, Mettler Toledo GmbH, Greifensee, Switzerland). The DSC tests were performed in a temperature range of 25–195 °C, using a heating and cooling rate of 10 °C/min and a nitrogen gas flow rate of 50 mL/min. Specimens with a mass ranging from 10 to 20 mg were encapsulated in standard aluminum hermetic pans and the heating–cooling–heating cycles were performed for all the samples. The degree of crystallinity of the polyethylene fraction was calculated by taking the reference enthalpy of fusion of pure HDPE crystals (293.6 J. g^−1^) [19] and corrected regarding the HDPE content according to Equation (1):(1)α=Δh/Δhc×100
where Δ*h* is the enthalpy fusion of the sample and Δ*hc* is the enthalpy of fusion of a 100% crystalline HDPE. Since the different compounds have different amounts of polyethylene, Δ*h* was normalized with the weight fraction of HDPE and or AAHDPE for each compound.

The experiment was repeated three times for each formulation to ensure the repeatability of the results.

#### 2.2.6. Rheological Characteristics of Compounds

The rheological properties of compounds A and B (Table 1) were evaluated using an oscillatory rotational rheology test in the rotational rheometer MCR 702 MultiDrive (Anton Paar GmbH, Graz, Austria), using parallel plates with a diameter of 25 mm, 2 mm height and angular frequencies from 0.1 to 500 rad/s using a strain value of 3%. To prevent the oxidation of the samples during experiments, nitrogen flow was used inside the chamber. After the melting, the samples were compressed to 1 mm by taking down the upper plate; and the test was started when the normal force felt down to zero. During the rotational rheology measurements of feedstocks and polymer-filled systems, a wall slip between the sample and the plates could be clearly observed. Therefore, the rheological properties of the filled polymers were examined in the high-pressure capillary rheometer Rheograph 2002 (Göttfert Werksto-ff-Prüfmaschinen GmbH, Buchen, Germany). The apparent shear viscosity was measured at 160 °C and the apparent shear rates from 75 to 2000 s^−1^; using a die of 30 mm in length and 1 mm in diameter. 

For both methods, at least three samples were tested for each type of binder or polymer-filled system; moreover, the average values are reported with standard deviation (±SD).

## 3. Results and Discussion

### 3.1. ATR

ATR spectroscopy was used to evaluate chemical changes in the different formulations. To investigate the homogeneity of samples, three different measurements were performed for each sample at three different points; no difference was observed between the measurements, indicating the homogeneity of the samples prepared from different formulations (data not shown). Figure 1 shows the comparison between the ATR-FTIR spectra of groups A, B, C, D and E. The presence of peaks at 1470, 2847, 2916 and 720 cm ^−1^ in the ATR spectra of samples containing HDPE and/or AAHDPE are related to in-plane vibrations of aliphatic hydrocarbons, symmetric and asymmetric stretching of aliphatic hydrocarbons and rocking of the same vibrational group in the structure of HDPE and AAHDPE, respectively [13,20].

The existence of the peak at 1714 cm^−1^ correlates to the stretching vibration of the carboxyl (C=O) of acrylic acid group in the formulations containing AAHDPE. The presence of a peak at around 2954 cm^−1^ in the FTIR spectra of samples containing PW corresponds to the presence of PW in the mixture according to previous literatures [21]. As PW is a mixture of hydrocarbon molecules containing 20–40 carbon atoms, the spectra of PW, HDPE and AAHDPE are very similar; in addition, no more special peak was distinguished in the ATR spectra of samples containing PW. The incidence of a peak at 1250 cm^−1^ in the C, D and E groups is related to presence of zirconia [13].

No new peak or significant peak shift was observed due to the formation of a chemical bond between the HDPE or AAHDPE with PW or zirconia. However, a slight shift of the carboxyl groups of the acrylic acid to lower wavelengths can be observed when comparing the formulations containing AAHDPE with and without zirconia; e.g., A100 and C100 (1714 to 1712 cm^−1^). The shift of the carboxyl groups peak can be attributed to the presence of hydrogen bonds between the hydroxyl groups in the zirconia surface and carboxyl groups, as observed in our previous study [13]. Moreover, the lower intensity of the zirconia peak at 1250 cm^−1^ in formulations containing AAHDPE than in those with only HDPE (C100 as compared to C0) can be a sign of a better covering of the powder surface by the polymer.

### 3.2. Morphology

Figure 2 and Figure 3 show the morphology of formulations C and E, respectively. As this study is focused on the effect of AAHDPE on the dispersion of the zirconia powder, the morphology of the unfilled formulations was not studied. Since the formulations D have the same components as formulations E, but with less powder content, the same trends were observed; thus, those images are not included here.

When comparing the morphology of formulations C with 30 vol% of zirconia and the different contents of HDPE and AAHDPE, clear differences can be observed (Figure 2). In C0, the HDPE is separated from the zirconia particles; in addition, polymer-rich areas can be observed, especially at higher magnifications. On the other hand, in C100, the zirconia powder is completely covered by the AAHDPE; furthermore, the mixture has a homogeneous microstructure. C50, with a mixture of HDPE and AAHDPE, shows a slightly better coverage of the particles than C0; however, it is far from the results of C100. The formation of hydrogen bonds observed with ATR (Section 3.1) and the low interfacial tension of AAHDPE with zirconia result in a strong powder-binder adhesion and in better powder dispersion and homogeneity for C100 [13].

Figure 3 shows the morphology of formulations E, the feedstocks of this study. As previously indicated, after many trials it was not possible to prepare E0, with only HDPE as polyethylene; thus, it is not included here. Comparing formulations C and E, it can be observed that the addition of PW results in an interconnected polymer network with small phases; in the case of E50 and C50, this is especially beneficial due to the improvement of the homogeneity and the apparition of smaller phases. Within formulations E, the increase in the AAHDPE content results in a reduction in the size of the polymeric phases; it also results in a more homogeneous microstructure, with a clear and gradual improvement from E25 to E100. This improvement could be mainly caused by the stronger polymer-powder adhesion with AAHDPE than with HDPE, which results in a better powder dispersion and homogeneity as observed for formulations C (Figure 2). A homogeneous feedstock microstructure with small polymeric phases is critical for the success of ceramic injection molding and similar processes; this is because it reduces the defects caused during the removal of the polymeric components in the debinding step [3,9]. More importantly, the improved powder-binder adhesion with AAHDPE could induce less powder-binder separation during the injection molding of complex components, when high shear forces are applied in the feedstocks [22].

### 3.3. DSC Analysis

DSC analysis was used to investigate the effect of including AAHDPE, PW and zirconia on crystallinity percentage, melting and crystallization temperature of the composites and feedstocks. Melting temperature determined using DSC analysis gives important information in the processing of polymers and their blends [23]. Figure 4 shows the representative melting (Figure 4a) and crystallization (Figure 4b) of the different types of formulations, using those with a 50/50 ratio of HDPE and AAHDPE as an example. DSC thermograms of samples containing only polymers (HDPE, AAHDPE or HDPE/AAHDPE) showed only one peak in the cooling and heating curves; while the existence of two distinct peaks in the DSC thermograms of samples containing PW corresponded to the presence of polymer(s) and PW. No additional peak was observed in the DSC thermograms of samples containing zirconia compared to those without zirconia (Figure 4).

The measured thermal properties of the different formulations listed in Table 1 are presented in Figure 5. As can be observed from Figure 5c, the presence of AAHDPE in the polymeric system decreases the crystallinity of HDPE; this is likely due to the branched structure of AAHDPE. It can be concluded that the HDPE chains are linear and can be aligned and packed together; while AAHDPE chains as branched polymer chains are hard to pack and lead to a decrease in crystallinity. Nevertheless, no noticeable influence on melting temperature was observed by increasing the AAHDPE content in the samples without zirconia (A0 to A100 and B0 to B100 in Figure 5a); a significant decrease in melting temperature by increasing AAHDPE was observed for the formulations containing zirconia (C0 to C100, D0 to D100 and E25 to E100 in Figure 5a).

The reduction of melting and crystallization temperatures was seen in the formulations containing PW compared to those without PW (group A vs. group C and group B vs. group D); this might be due to the plasticizing effect of PW. However, the comparison between the samples with and without PW (A vs. B and C vs. D) clearly indicates that the presence of PW does not have noticeable effect on the crystallinity of the samples. Matula et al. also reported that paraffin significantly reduces the melting point of polypropylene and HDPE [23].

As shown, the formulations containing zirconia have higher melting temperatures compared to those without zirconia; further, these differences are more significant for formulations without PW (group A vs. C). The crystallization temperature decreased by including zirconia into the structure of binders without PW, meaning the delay of crystallization by including zirconia in the structure of binders; this is likely due to the higher distance between polymeric chains in the presence of zirconia (group A vs. group C). For formulations containing PW (B vs. D), the reduction in the crystallization temperature was more remarkable for the combinations containing AAHDPE. No noticeable difference between the melting and crystallization temperature was observed for the formulations containing 30 vol% and 50 vol% zirconia (D vs. E).

### 3.4. Rheological Investigation

Understanding the rheological properties is important as it gives us information regarding the internal structure of materials; and an estimation of the processing conditions for practical polymer processing, such as injection molding. Viscosity is the most important property of filled-polymeric systems, which is measured using either capillary or rotational rheometric methods [24,25]. Although rotational rheology is carried out at relatively low shear rates, it provides a fundamental overview of the structure and interactions of the components of the mixture [25]. During this study, rotational rheology was used to evaluate the rheological properties of binders with and without PW (Groups A and B). Due to the wall slip effect of filled-polymers’ discs between upper and lower plates of the rotational rheometer observed during measurement, the rheological properties of filled-polymers containing 30 vol% and 50 vol% zirconia were investigated with a capillary rheometer.

Pseudoplastic behavior was observed for A0, A50 and A100 in the range of angular frequencies assessed during this study (Figure 6a). Pseudoplastic behavior has also been observed in previous studies for HDPE and branched polyethylene or grafted polyethylene with polar groups [26,27,28]. The disentanglement of polymeric chains is affected by the chemical structure of polymers. As can be observed from Figure 6a, the complex viscosity of A100 is higher than that for A0 and A50, especially at lower angular frequencies. It can be attributed to hydrogen bonds between the acrylic acid groups in the structure of AAHDPE and the higher interaction of AAHDPE polymeric chains at lower angular frequencies; thus, this results in more effective interaction and entanglement [13]. Branching and partial crosslinking of the AAHDPE polymeric chains might be another reason for the higher complex viscosity of the A100 samples. The higher viscosity of grafted polyethylene compared to that for the un-grafted one has been reported in previous studies [13,28]. However, at higher angular frequencies, these interactions break down and the difference between the viscosities of A100, A50 and A0 is less significant. There are many entanglements between the HDPE polymeric chains due to the flexible nature of HDPE chains, especially at lower angular frequencies. However, their conformations can be changed and disentangled by increasing the angular frequency.

The complex viscosity of A50 was found to be much lower compared to that for A0 and A100. Due to the presence of HDPE and AAHDPE in the structure of A50, the hydrogen bonds between AAHDPE might not be as strong as A100 since there are HDPE chains between AAHDPE chains. Moreover, the entanglements between HDPE chains are not as strong as in A0 as there are AAHDPE chains in the space of HDPE chains. As A50 likely has lower amounts of entanglements and hydrogen bonds between polymeric chains compared to A0 and A100, it could result in a significantly lower complex viscosity for A50.

The storage and loss modulus indicate the elastic and viscous behavior of materials, respectively [29]. The same trend as for complex viscosity was observed for the storage and loss moduli for A0, A50 and A100. The lowest storage and loss moduli were observed for A50 compared to A0 and A100; this can be attributed to a weaker polymeric chain network formation and a lower interaction between polymeric chains in the A50 formulation. In angular frequencies above 10 rad/s, there is no visible difference between the storage modulus of different samples; this might be attributed to the breakdown of the network and the reduction of entanglements between polymeric chains at these angular frequencies; while the higher interaction and entanglements between polymeric chains in the lower angular frequencies make a noticeable difference between the storage modulus of different samples. The higher the interaction and entanglements between polymeric chains, the higher the storage and loss moduli.

The loss factor or tan delta is defined as the ratio of loss to storage modulus; it is a helpful parameter to indicate the relation between the elastic and viscous fractions of viscoelastic materials. As can be seen in Figure 6d, A100 has the lowest loss factor compared to A50 and A0 indicating the more elastic behavior of A100; while A0 has the highest loss factor revealing the more viscous behavior of this sample. The loss factor values of A50 were between those of A100 and A0; they were closer to A0 at low angular frequencies, but with a pronounced slope. Noticeable differences in the viscoelastic properties at low angular frequencies for A0, A50 and A100 can be attributed to the chain entanglement or crosslinking and hydrogen bonds between the acrylic acid groups in AAHDPE [13].

The complex viscosities of the binders containing PW (group B) were found to be significantly lower than those for the binders without PW (group A); this can be attributed to the low viscosity of PW [10]. PW has a much lower molecular weight compared to the polymers; therefore, it has a substantially lower viscosity (around 2.8–6.8 mPa.s depending on the temperature) than polymers such as HDPE and AAHDPE at the processing temperature [30]. The decrease in the viscosity of the binders with the addition of paraffin has been reported in a previous study by Matula et al. [23]. At small angular frequencies, the viscosity of B100 was slightly higher than that of B0; this follows the same trend as the viscosity of A100 and A0. However, a different trend was observed at higher angular frequencies, where the complex viscosity of B100 was found to be lower than that for B50 and B0. Moreover, B100 shows a pseudoplastic behavior in the range of angular frequencies; whereas a Newtonian plateau at low angular frequencies was observed for B50 and B0. It is likely due to the presence of PW molecules between AAHDPE chains; this increases the distance between the polymeric chains and results in lower amounts of hydrogen bonds between the acrylic acid groups. In addition, due to the larger distance between polymeric chains in the presence of PW, these hydrogen bonds can easily break down even at a lower angular frequency. The Newtonian plateau for the mixtures containing HDPE might be due to the presence of high amounts of entanglements between chains, which prevents the flow of materials. As PW has a similar chemical structure and good compatibility with HDPE [3], the presence of PW might not affect the chain entanglement at low angular frequencies.

A behavior similar to the complex viscosity was observed for the storage and loss moduli of B0, B50 and B100. However, the difference between the loss modulus of the samples containing PW is not noticeable, especially at angular frequencies less than 1 rad·s^−1^. As illustrated in Figure 6d, B0 has the highest loss factor, especially at low angular frequencies; this is followed by B50 in the intermediate and close to B0 at high angular frequencies; and B100 has the lowest loss factor. As the higher interaction between polymeric chains results in a higher loss factor [31], the highest loss factor for B0 might be due to the high amounts of entanglements between the HDPE polymeric chains and between the HDPE polymeric chains and PW chains with a similar chemical structure.

Capillary rheometry can mimic high shear environments that usually occur in injection molding processes. Figure 7 illustrates the results of apparent viscosity measurements at different apparent shear rates of 75 to 2000 s^−1^ for the C, D and E groups. Pseudoplastic behavior was observed for all of the samples. No noticeable difference was observed for the viscosity of C0 and C100; while the viscosity of C50 was found to be the lowest one in the evaluated apparent shear rates range (Figure 7a). Considering the clear differences between the microstructure in formulations C (Figure 2), a lower apparent viscosity could be expected for C100 than for C0 due to the better powder-binder adhesion and powder dispersion in C100 than in C0. However, as observed in our previous study with similar formulations, the higher adhesion of AAHDPE to the powder and the improvement of the powder dispersion results in an increase in the apparent viscosity due to the formation of a powder-binder network [13]. The fraction of AAHDPE in C50 results in an increase in the powder-binder adhesion and in a reduction in the agglomerates compared to C0, with only HDPE. However, the interaction between HDPE and AAHDPE results in a considerable decrease in the polymer viscosity as observed for A50 in Figure 6a.

The viscosity of the D group (Figure 7b) was found to be lower than that for C (Figure 7a); this is due to the presence of PW and the lower viscosity of the binder in group D than in group C and is consistent with previous reports [23]. The higher viscosity of D0 compared to D50 and D100 was observed (Figure 7b); this can be related to the poor interaction between zirconia and HDPE and the possible agglomeration of zirconia. Moreover, many pressure oscillations occurred during the measurements at low apparent shear rates for D0; this might be caused by the powder agglomerates. Because of these pressure oscillations, the apparent viscosity values of D0 at low apparent shear rates have been discarded from Figure 7b. However, at higher apparent shear rates, the difference between the apparent viscosity of D0 with D50 and D100 is lower; this is likely due to the breakage of zirconia agglomeration at a higher shear rate. No significant difference was observed between the viscosity of D50 and D100, revealing that the addition of 25 vol% of AAHDPE in a polymeric mixture filled with 30 vol% of zirconia powder prevents the powder agglomeration and the viscosity increment.

As expected [29,32], the increase in powder content from 30 vol% in group D to 50 vol% in group E increases the apparent viscosity of all the formulations. As previously indicated, E0 could not be prepared due to the poor interaction of ceramic powder and HDPE; thus, the rheological properties of E0 could not be measured. Moreover, the high apparent viscosity of E25 compared to E50, E75 and E100 (Figure 7c) indicates that 12.5% of AAHDPE in the binder still cannot prevent the agglomeration of zirconia in a feedstock with 50 vol% of powder. The poor powder dispersion in E25 could be clearly observed during the rheology measurements: at low apparent shear rates, intervals of no material flow intercalated with others of sudden flow of material periodically; and after each shear rate increase at intermediate and high apparent shear rates, a rapid pressure increase occurred, followed by the pressure reduction and stabilization. Minor pressure variations could be observed for E50, E75 and E100, which were of less magnitude as the AAHDPE content in the feedstock increased. In fact, E100 only had minor pressure variations in one of the repetitions. As can be observed from Figure 7c, increasing the ratio of AAHDPE decreased the apparent viscosity of feedstocks; this can be attributed to better feedstock homogeneity and dispersion of zirconia in feedstocks containing more amounts of AAHDPE (Section 3.2). The viscosity decrease is especially noticeable from E25 to E50 and minor from E50 to E75; there is almost no difference from E75 to E100. These results indicate that for the feedstocks with 50 vol% of the zirconia feedstock evaluated here, the critical concentration of AAHDPE, above which there is no further improvement of properties with the increase in the compatibilizer or dispersant, is in between those of E75 and E100 [33,34].

## 4. Conclusions

The effect of the addition of AAHDPE on the properties of feedstocks for the ceramic injection molding of zirconia was studied. Relevant peaks related to HDPE, AAHDPE zirconia and PW were observed in the ATR spectra of different formulations. The ATR spectra confirmed the presence of hydrogen bonds between the hydroxyl groups of zirconia and the carboxyl groups of acrylic acid. The hydrogen bonds and the low interfacial tension between AAHDPE improve the adhesion between the powder and binder; this results in a better dispersion of zirconia and more homogeneity of formulations containing higher amounts of AAHDPE, as observed in the SEM images. The results of DSC showed the decrease in melting temperature by increasing AAHDPE for the formulations containing zirconia. Similarly to previous studies, the reduction of the melting and crystallization temperatures were noticed in formulations containing PW compared to those without PW. The interaction between the binder components and the adhesion with the powder determine the rheological properties of the different formulations. It can be concluded that the difference in the complex viscosity of different formulations of groups A and B might be related to the branching, partial crosslinking and or hydrogen bonding between the poly (acrylic acid) presented in the structure of AAHDPE, the entanglements between the polyethylene chains and the interaction between PW and the polyethylene molecules. For the formulations containing zirconia (groups C, D and E), the incorporation of AAHDPE is needed to guarantee a continuous flow and low viscosity. In fact, the increase in the AAHDPE content facilitates the incorporation in the feedstocks. The results of this study provide a better insight into the powder-binder interaction in ceramic injection molding feedstocks; in addition, they show the potential of grafted polymers as binders for CIM and similar processes based on polymer systems highly filled with ceramic powder.

## Figures and Tables

**Figure 1 polymers-14-03653-f001:**
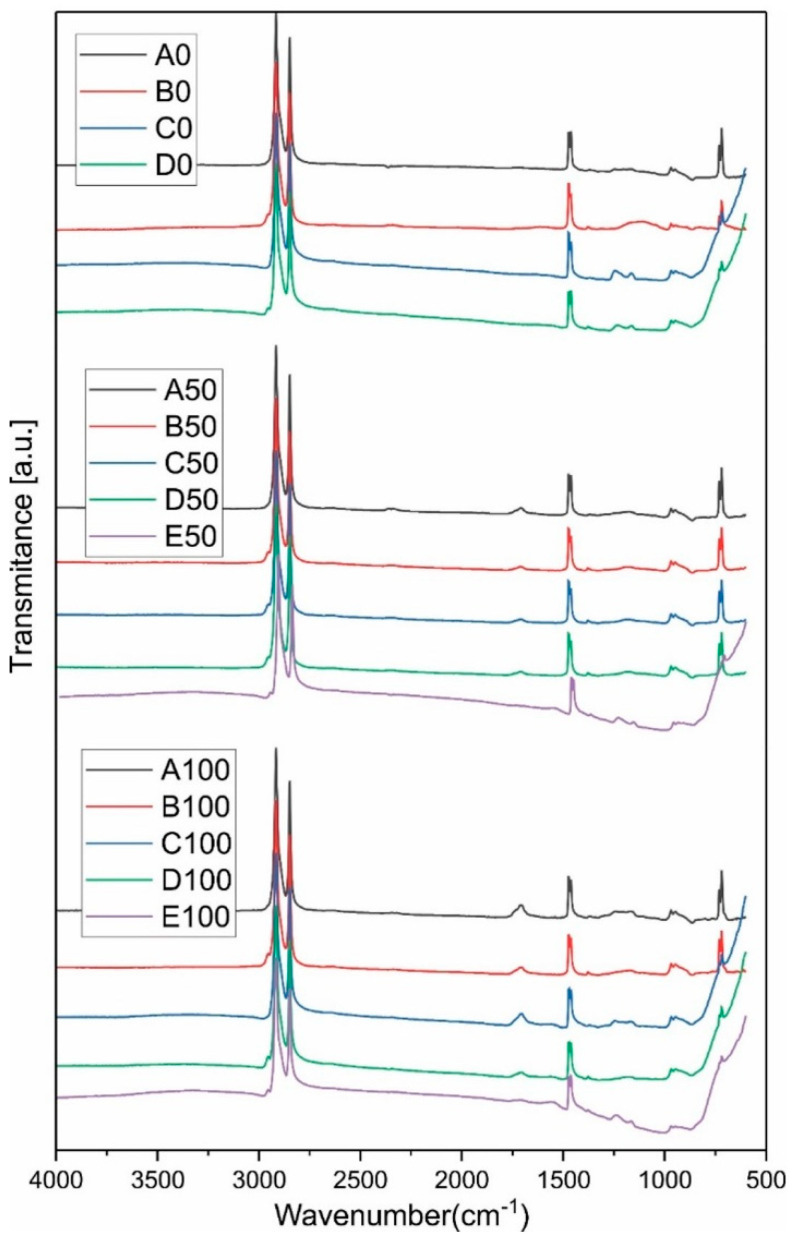
Comparison between the ATR-FTIR spectra of different samples.

**Figure 2 polymers-14-03653-f002:**
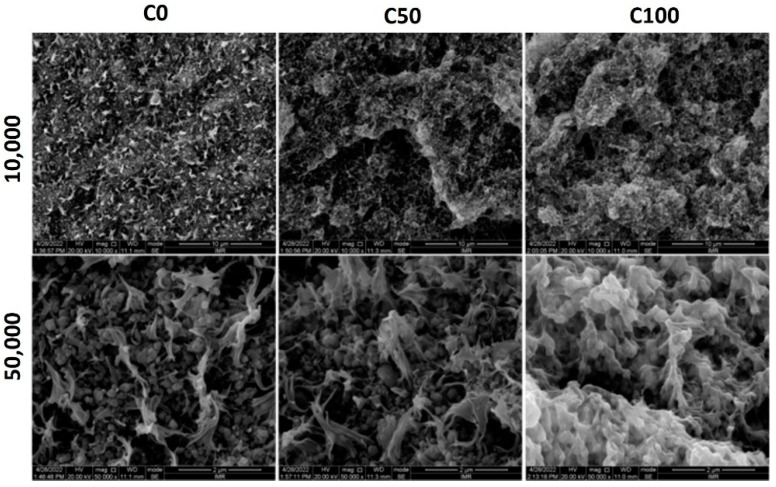
Morphology of formulations C at 10,000 and 50,000 magnifications.

**Figure 3 polymers-14-03653-f003:**
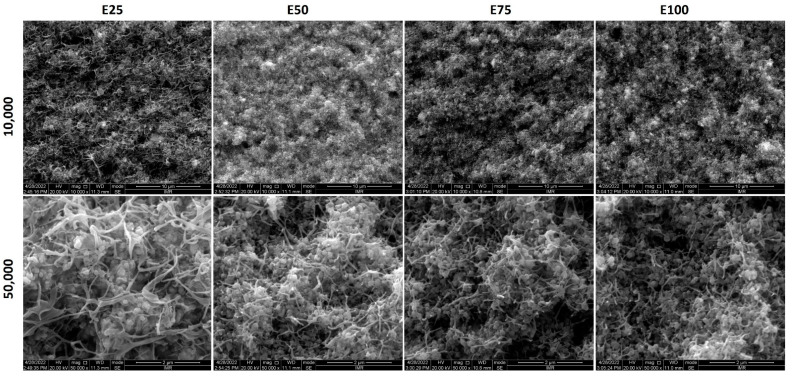
Morphology of formulations E at 10,000 and 50,000 magnifications.

**Figure 4 polymers-14-03653-f004:**
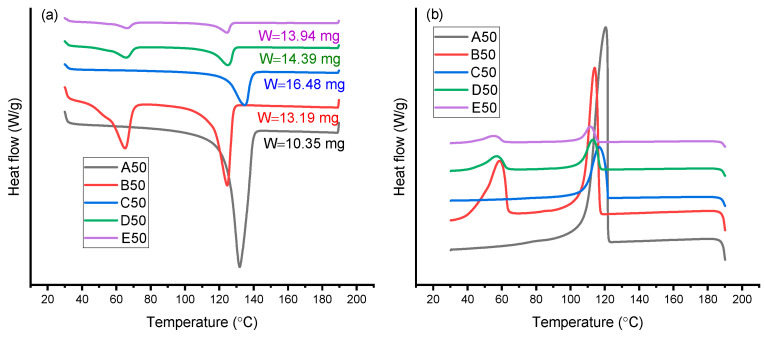
Representative (**a**) heating and (**b**) cooling DSC curves of A50, B50, C50, D50 and E50.

**Figure 5 polymers-14-03653-f005:**
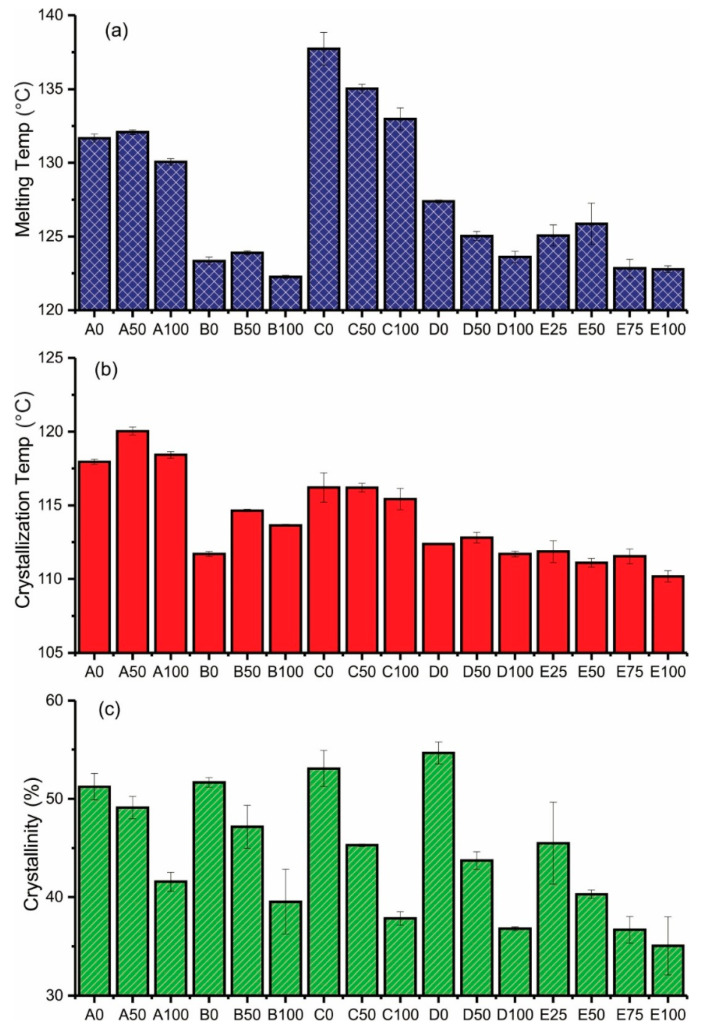
(**a**) Melting temperature (°C), (**b**) crystallization temperature (°C) and (**c**) crystallinity (%) of polyethylene in the different samples.

**Figure 6 polymers-14-03653-f006:**
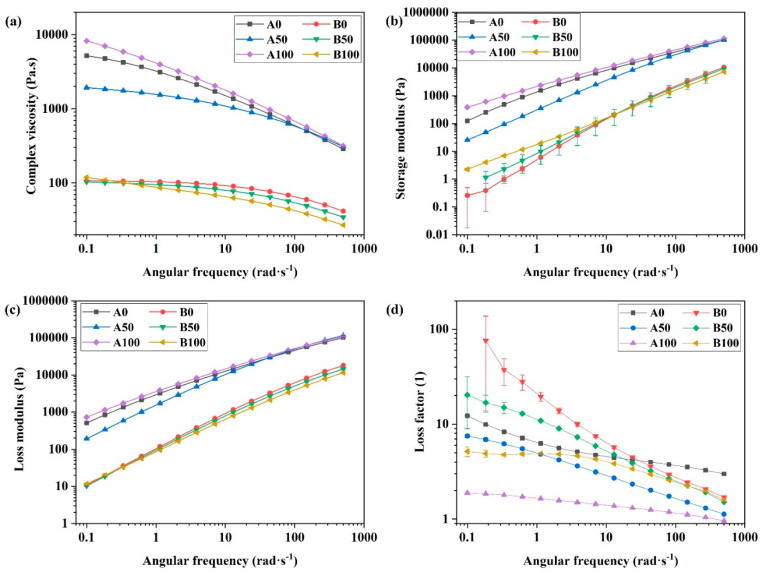
Rheological properties (viscosity (**a**), storage modulus (**b**), loss modulus (**c**) and loss factor (**d**)) of the A and B groups at different angular frequencies.

**Figure 7 polymers-14-03653-f007:**
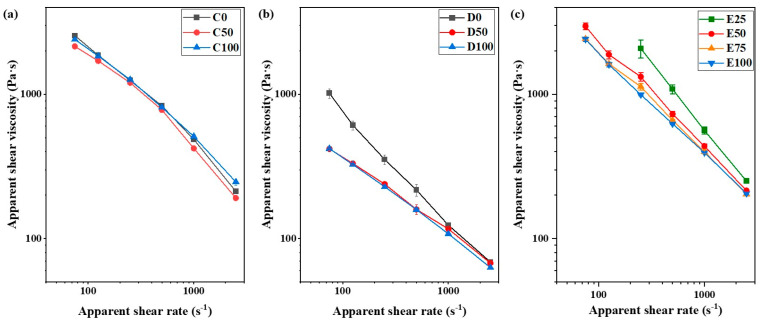
Apparent shear viscosity at different apparent shear rates of the different formulations of groups: (**a**) C, (**b**) D and (**c**) E.

**Table 1 polymers-14-03653-t001:** Volumetric composition of the different compounds designed for this study.

	Binder Composition (vol%)			Powder Content (vol%)
	HDPE	AAHDPE	PW	
A0	100			
A50	50	50		
A100		100		
Ax			100	
B0	50		50	
B50	25	25	50	
B100		50	50	
C0	100			30
C50	50	50		30
C100		100		30
Cx			100	30
D0	50		50	30
D50	25	25	50	30
D100		50	50	30
E0	50		50	50
E25	37.5	12.5	50	50
E50	25	25	50	50
E75	12.5	37.5	50	50
E100		50	50	50

**Table 2 polymers-14-03653-t002:** Different stages of vacuum press compression molding.

Stage (No)	Temperature (°C)	Pressure (bar)	Time (min)
1	160	0	10
2	160	50	5
3	30	50	10

## Data Availability

The data presented in this study are available on request from the corresponding author.

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
