# Peer review of "Effect of Increased Powder–Binder Adhesion by Backbone Grafting on the Properties of Feedstocks for Ceramic Injection Molding"

_polymers, 2022, doi:10.3390/polym14173653_

Round 1

Reviewer 1 Report

Authors presented a study on the additives to binder particles to increase their adhesion.

The work is voluminous and has been constructed well. The manuscript is well written with sufficient information and discussions.

A few comments before this work can be published

1. Authors speak greatly about adhesion, however, there are no investigations concerning the wettability/adhesion. It is more of a qualitative approach but it would be nice to have something quantitative. I agree that ATR does provide insights but wettability measurements also do add value to the work.

2. Did you also consider the weight fractions of additives while calculating the crystallinity within the samples?

Reviewer 2 Report

In this manuscript, the rheological, thermal and chemical properties of polymer composites highly-filled with zirconia and feedstocks were evaluated. This work was well organized. However, the following issues should be addressed by authors.  

1.     The main conclusion is not clear in this manuscript. As stated “There are only few studies focused on the effect of binder formula on feedstock structure. Therefore, the selection of the binder ingredients and their careful balancing needs to be optimized.” How did the binder ingredients influence the feedstock structure?

2.     It is suggested that the authors should give the schematic of the interactions between HDPE, AAHDPE and zirconia particles.

3.     As the author stated “ATR spectra confirmed the presence of hydrogen bonds between zirconia and carboxyl groups” How did the authors draw this conclusion only based on a negligible shift in ATR spectra.

4.     Please revise “FTIR” as “ATR” or ATR-FTIR in the caption of Figure 1.

5.     Please check the name (unit) of vertical axis in Figure 5a, b.

Round 2

Reviewer 2 Report

This manuscript could be accepted in present version.